# The genetic evolution of acral melanoma

Meng Wang [1,2], Satoshi Fukushima[3], Yi-Shuan Sheen[4], Egle Ramelyte [5], Noel Cruz-Pacheco [1,2], Chenxu Shi[1,2], Shanshan Liu[1,2], Ishani Banik[1,2], Jamie D. Aquino[6], Martin Sangueza Acosta[7], Mitchell Levesque [5], Reinhard Dummer [5], Jau-Yu Liau[4], Chia-Yu Chu[4], A. Hunter Shain [1,2,8], Iwei Yeh [1,2,6,8] & Boris C. Bastian [1,2,6,8] ✉

Acral melanoma is an aggressive type of melanoma with unknown origins. It is the most common type of melanoma in individuals with dark skin and is notoriously challenging to treat. We examine exome sequencing data of 139 tissue samples, spanning different progression stages, from 37 patients. We find that 78.4% of the melanomas display clustered copy number transitions with focal amplifications, recurring predominantly on chromosomes 5, 11, 12, and 22. These complex genomic aberrations are typically shared across all progression stages of individual patients. *TERT* activating alterations also arise early, whereas MAP-kinase pathway mutations appear later, an inverted order compared to the canonical evolution. The punctuated formation of complex aberrations and early *TERT* activation suggest a unique mutational mechanism that initiates acral melanoma. The marked intratumoral heterogeneity, especially concerning MAP-kinase pathway mutations, may partly explain the limited success of therapies for this melanoma subtype.

UV radiation is the major mutational mechanism for most melanomas on the skin, increasing its prevalence in populations with light skin tone and poor tanning ability. A subset of melanomas called acral melanomas arise on the palms, soles, and nail apparatus—sites which receive little UV radiation and are further shielded from the sun by a thick cornified layer or nail plate. They typically do not arise from melanocytic nevi[1], benign precursors of melanomas on the sun-exposed skin. The relative incidence of acral melanomas among Caucasians is low compared to UV radiation-induced melanomas. However, their absolute incidence is comparable across world populations, and acral melanoma represents the major melanoma subtype in populations with darker skin tones. Acral melanomas have a more aggressive course with greater mortality than cutaneous melanomas and respond less well to immune-based and targeted therapies[2–4].

Concordant with their anatomic distribution, the genomes of acral melanomas have a low burden of point mutations and mostly lack UV-radiation-induced mutational signatures. These features indicate that they are not caused by UV radiation but other, yet-to-be-identified, mutational mechanisms. The relative dearth of point mutations in acral melanoma is contrasted by a high burden of copy number and structural aberrations, in particular the presence of multiple narrow, high-level amplifications, deep deletions, and complex intra- and inter-chromosomal rearrangements[5–7]. Large-scale integrative analyses from the Pan-Cancer Analysis of Whole Genomes Consortium[8] confirmed a high frequency of distinctive complex rearrangements including chromothripsis and chromoplexy (tyfonas)[9] in acral melanomas that distinguished them from most other cancer types. Together, these findings suggest a unique form of genomic instability in melanomas originating from acral sites.

The spectrum of driver mutations in acral melanomas also diverges from melanomas on the sun-exposed skin. Only a minor fraction of acral melanomas bear point mutations in *BRAF* and *TERT*,

---

[1]Department of Dermatology, University of California San Francisco, San Francisco, CA, USA. [2]Helen Diller Family Comprehensive Cancer Center, University of California San Francisco, San Francisco, CA, USA. [3]Department of Dermatology and Plastic Surgery, Faculty of Life Sciences, Kumamoto University, Kumamoto, Japan. [4]Department of Dermatology, National Taiwan University Hospital and National Taiwan University College of Medicine, Taipei, Taiwan. [5]Department of Dermatology, University of Zurich, Zurich, Switzerland. [6]Department of Pathology, University of California San Francisco, San Francisco, CA, USA. [7]Hospital Obrero, Caja Nacional de Salud, La Paz, Bolivia. [8]These authors jointly supervised this work: A. Hunter Shain, Iwei Yeh, Boris C. Bastian. ✉e-mail: boris.bastian@ucsf.edu

which are common in sun-exposed melanomas. By contrast, acral melanomas are more often driven by high amplitude amplifications of *CCND1*, *CDK4*, *MDM2*, and *TERT*, among other genes. The timing of mutations has been studied in great detail in the setting of sun-exposed melanomas[10–12]. MAP-kinase pathway-activating mutations undergo early selection during the evolution of melanomas on sun-exposed skin, followed by mutations impairing other pathways such as telomere maintenance and cell-cycle checkpoint control. The order in which mutations undergo selection during the evolution of acral melanoma is currently not known.

Understanding the order in which somatic alterations undergo selection as well as the mutational mechanisms operating at each phase of tumorigenesis will have clinical relevance. For instance, a sustained form of genomic instability would point towards an underlying DNA repair defect, which, once characterized, could likely be therapeutically exploited with synthetic lethal approaches of complementing pathways, analogous to BRCA or mismatch repair deficient cancers[13,14].

In this study, we compare different progression stages of acral melanomas in thirty-seven patients to show that the characteristic complex copy number and structural rearrangements arise early and in a punctuated rather than ongoing pattern. In addition, we show that mutational activation of telomere maintenances mechanisms precedes the emergence of MAP-kinase pathway mutations, implicating telomere maintenance in re-stabilizing genomes after a phase of instability.

## Results

We sequenced the exomes of two or more tumor areas representing different progression stages, including the in situ and invasive portions of the primary tumor and available regional or distant metastases, with patients' corresponding normal tissues from thirty-five acral melanomas. To this cohort, we added two acral melanomas from which sequencing data from two different progression stages and normal tissues existed from previous studies[15,16]. In total, we obtained sequencing data from one hundred thirty-nine tissue samples (Supplementary Fig. 1 and Supplementary Data 1) with an average unique coverage of 120-fold for tumor tissues and 77-fold for normal tissues (Supplementary Data 1).

### Acral melanomas are enriched for complex copy number aberrations with characteristic patterns

For each melanoma from our cohort, we identified and annotated germline and somatic mutations and copy number alterations, estimated tumor purity and ploidy, and inferred the phylogenetic relationship of tumor samples [Supplementary Data 1–3; Supplementary Fig. 2 and available on Figshare[17]]. From the copy number profiles we identified high-level amplifications, defined as segments with copy numbers exceeding background tumor ploidy by a factor of three or more, in 81.1% (30 of 37) of cases. We noted that most of these amplifications were flanked by clusters of copy number transitions (CNTs) resulting in numerous neighboring segments with varying copy numbers (Fig. 1a). Analogous to the concept of kataegis, which is defined as regions with increased density of point mutations, we refer to these regional clusters of CNTs including high-level amplifications (details see Materials and Methods) as hailstorms (Greek chalazothýella). We note that many of these hailstorms would likely qualify as "tyfonas", a term coined using structural information from whole genome sequencing data[9]. In our cohort of 37 cases, 29 (78.4%) harbored one (8 cases or 21.6%) or multiple (21 cases or 56.8%) hailstorms, with a maximum of five hailstorms per case (3 cases or 8%). For comparison, only 27 of 358 (7.5%) of cutaneous melanomas in TCGA harbored hailstorms ($P < 2.2 \times 10^{-16}$, Fisher's Exact Test), with only 7 cases (2%) having more than one hailstorms.

The genomic regions affected by hailstorms were not randomly distributed across the genome but frequently involved chromosomes 5p (including *TERT*, *RICTOR* and *SKP2*), 11q (including *CCND1*, *PAK1*, *GAB2* and *YAP1*), 12q (including *CDK4* and *MDM2*) and 22q (including *CRKL*, *SOX10* and *EP300*) (Fig. 1b and Supplementary Fig. 2), in line with prior genetic analyses of acral melanoma[6,7,18,19]. The non-random distribution of hailstorms with amplifications recurrently targeting known melanoma oncogenes indicates that these alterations are under positive selection. The hailstorms were accompanied by foci of kataegis (Fig. 1a) in 62.1% (18/29) of cases.

### Hailstorms tend to arise early during the evolution of acral melanoma

We found that 92.2% (59/64) of the hailstorms, in 28 of the 29 (97%) melanomas in which they were present, were already identifiable in the earliest in situ progression stages and shared a nearly identical pattern with tumor areas representing later stages (Fig. 1c and Supplementary Figs. 2, 3). Only five hailstorms (7.8%) were acquired later during progression and private to some of the areas. The example case in Fig. 1d displays three different hailstorms affecting chromosomes 5, 11, and 19. The pattern of copy number changes and genomic coordinates of CNTs are highly similar between the in situ and two different invasive portions, indicating that they arose at or before the in situ stage. Whole-genome sequencing of this tumor (Supplementary Fig. 4) confirmed the highly concordant copy number patterns across the different tumor areas and in addition revealed numerous shared structural variation (SV) junctions within hailstorms, confirming their clonal origin. Some of these SV junctions were also identifiable in the exome sequencing data (Supplementary Fig. 5). Together, these findings indicate that hailstorms arose in a punctuated fashion and were generated by mechanisms that operated before or during the in situ stage but largely abated as these tumors progressed further.

### *TERT* activation is the most common genetic alteration in acral melanoma and arises early during progression

The punctuated nature of the emergence of hailstorms suggests that they may result from double-stranded DNA breaks that occur early during tumor evolution. These breaks could be due to physical force implicated in acral melanoma due to its preference for pressure-bearing sites of the feet[20,21] or telomere erosion during crisis[22,23]. Both can generate breakage-fusion-bridge cycles that can trigger complex chromosomal rearrangements and chromothripsis, the shattering of large chromosomal regions or entire chromosomes.

In our cohort, *TERT* stood out as the most frequently altered gene with 26 of 37 (70.3%) melanomas harboring *TERT* alterations (Supplementary Fig. 2). Most commonly, *TERT* was amplified as part of a hailstorm on chromosome 5p, but *TERT* promoter mutations and CNTs immediately upstream of *TERT*, which are implicated in *TERT* activation[24], were also recurrently identified (Fig. 2a, b). All *TERT* alterations were present at the earliest progression stages, placing them on the trunk of the phylogenetic trees of all tumors in which they were present (Fig. 2). *ATRX* and *DAXX* alterations, genetic alterations associated with alternative telomere lengthening (ALT), were not observed in our cohort. The punctuated emergence of hailstorms and early somatic activation of telomere maintenance would support the notion that telomere crisis could be a mutational mechanism in acral melanoma that acts early on in their formation. Of note, telomerase can also 'heal' chromosomal breaks outside of telomeres[25], presumably also those caused by physical trauma to cells.

### UV signature mutations arise after initiation in acral melanoma with UV-induced DNA damage

A UV radiation signature, almost universally present in cutaneous melanomas, has been reported in a few instances of acral melanoma[7,19]. In our cohort, five melanomas showed evidence of mutational signature SBS7, characteristic of UV radiation. Interestingly in two of the

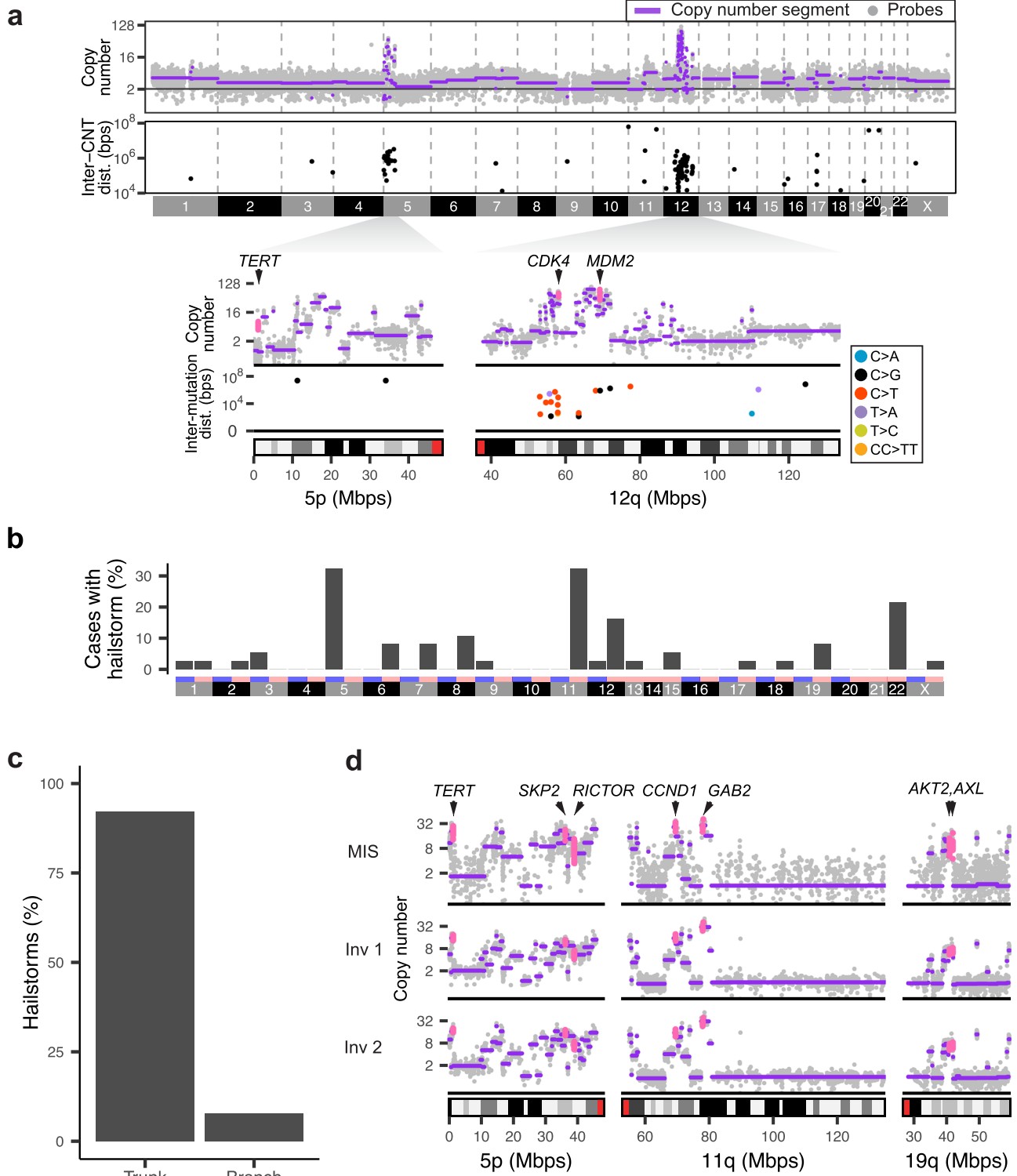

**Fig. 1 | Clusters of copy number transitions (hailstorms) are common in acral melanomas and arise early in their evolution. a** A melanoma (case 101) with hailstorms on chromosome 5p and 12q (upper panel) defined as genomic regions with high-level amplifications and high density of copy number transitions (CNT) (lower panel). The higher resolution panels underneath show the location of putative driver genes within the amplicons and reveal foci of kataegis on 12q. **b** Hailstorms in acral melanoma are distributed non-randomly, and preferentially involve chromosomes 5, 11, 12, and 22. **c** Most hailstorms are shared across all samples of a given patient, placing them on the trunk of the respective phylogenetic tree. **d** An example case (case 110) with hailstorms on 5p, 11q, and 19q shared across the melanoma in situ (MIS) and two separate invasive areas (Inv 1 and Inv 2) of the primary melanoma. The copy number profiles of all three tumor areas show identical hailstorms on three chromosomal arms with congruent copy number transitions. The coding regions of amplified oncogenes are highlighted in pink. Source data are provided as a Source Data file.

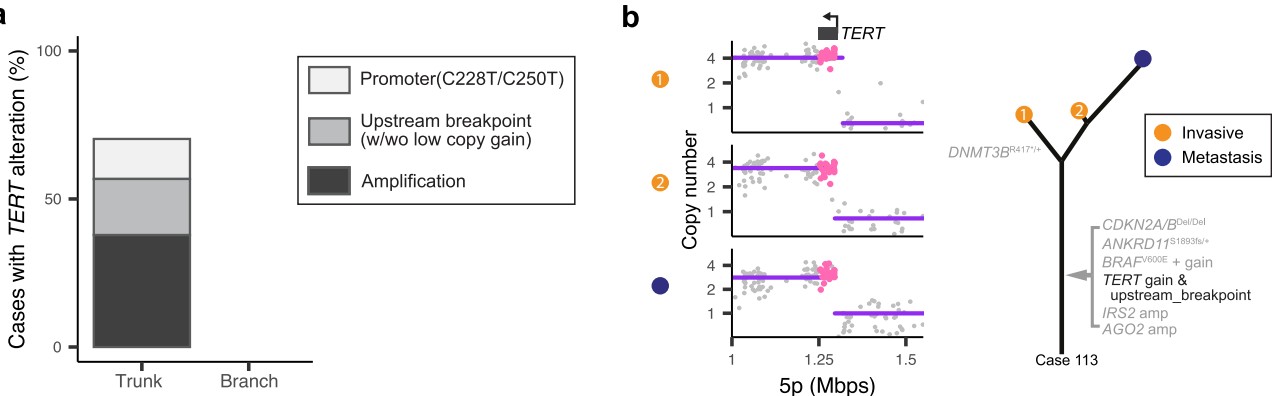

**Fig. 2 | *TERT* alterations arise during the earliest progression stages.**
**a** Amplifications, upstream structural rearrangements, and promoter mutations of *TERT* all occurred on the trunk of phylogenetic trees and invariably were identifiable in the earliest progression stage of melanomas that carried these alterations. The relative frequency of the different types of *TERT* alterations are shown. **b** An example case with structural rearrangement immediately upstream of *TERT* (left) shared across all three tumor areas from two different progression stages. The corresponding phylogenetic tree is shown (right). Source data are provided as a Source Data file.

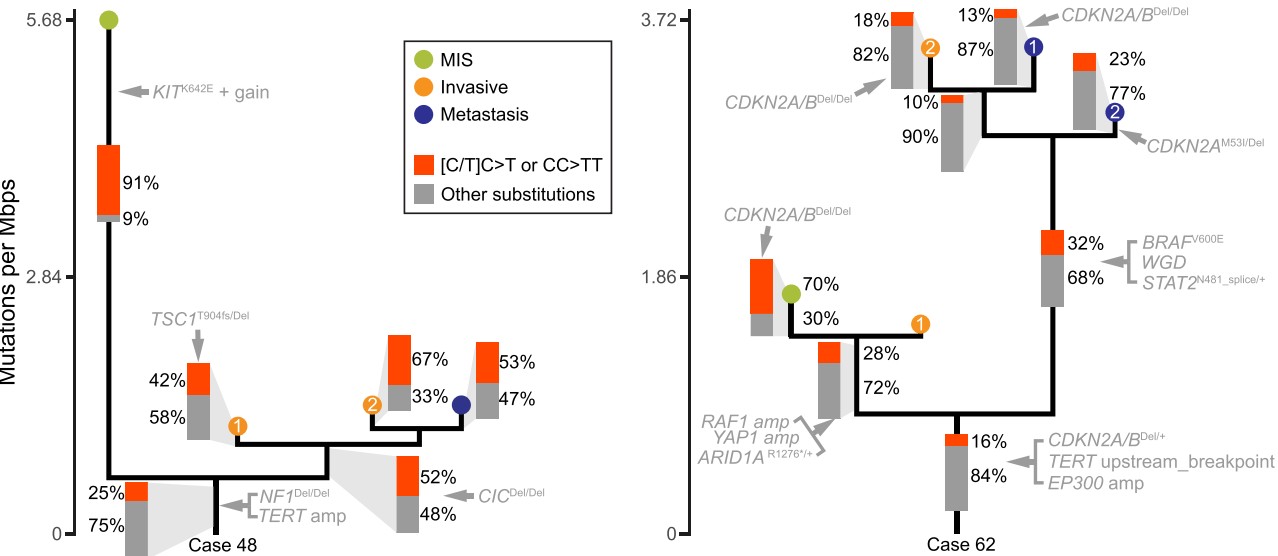

**Fig. 3 | Two acral melanomas with UV signature mutations accumulating after tumor initiation.** For both cases, the highest percentages of [C/T]C > T or CC > TT substitutions were observed in the MIS areas, which in case 48 (left panel) had an elevated mutation burden (5.7 vs. 1.1–1.3 mutations/megabase). The fractions were lower on the evolutionary trunks of both melanomas, suggesting limited or no exposure to UV radiation in the most recent common ancestors of both tumors. Source data are provided as a Source Data file.

five melanomas, the UV signature was detectable mostly in the in situ areas but was nearly absent in the more advanced progression stages (Fig. 3 and Supplementary Fig. 6). This suggests that in acral melanoma with exposure to the sun, UV radiation mainly acted *after* the initiation of the neoplastic growth and therefore likely was not an initiating mutational mechanism but can increase intratumoral heterogeneity.

Other mutational signatures identified were SBS3, attributed to defective homologous recombination repair and observed in 9 melanomas, as well as SBS2 and SBS13, attributed to APOBEC (Supplementary Fig. 6) in seven cases. Hailstorm-associated kataegis was shared across different progression stages and thus also arose early, likely at the formation of the hailstorms.

### Mutations activating the MAP-kinase pathway often emerge after tumor initiation, sometimes resulting in competing subclones driven by different oncogenes

Point mutations that activate the MAP-kinase pathway are the earliest genetic alterations arising during the progression of cutaneous and uveal melanomas[10,12,26,27]. Surprisingly, we found multiple acral melanomas in which MAP-kinase pathway mutations emerged only later during progression (Figs. 4 and 5 and Supplementary Fig. 2). In some cases, spatially separated tumor cell populations of the same primary tumor harbored different MAP-kinase pathway driver mutations. For example, in case 62 a *BRAF* V600E mutation was present at near clonal level in one part of the invasive component of the primary tumor (area B in Fig. 4a) and all three areas of two different metastases, but only identified in trace amounts in an adjacent area of the invasive component of primary and the in situ melanoma flanking (area A and MIS in Fig. 4a). Area A and the melanoma in situ instead harbored amplifications of *RAF1 (encoding *CRAF*) as the likely MAP-kinase pathway driver (Fig. 4a–c), and this *RAF1* amplification was absent from all other areas. Early phylogenetic separation of these genetically different tumor cell subpopulations was further supported by a *STAT2* mutation as well as whole genome doubling (WGD) exclusive to area B and the metastases, while area A and the in situ area instead harbored a heterozygous truncating *ARID1A* mutation and a *YAP1* amplification (Fig. 4b, c). Immunohistochemistry of YAP1 with strong immunoreactivity of areas

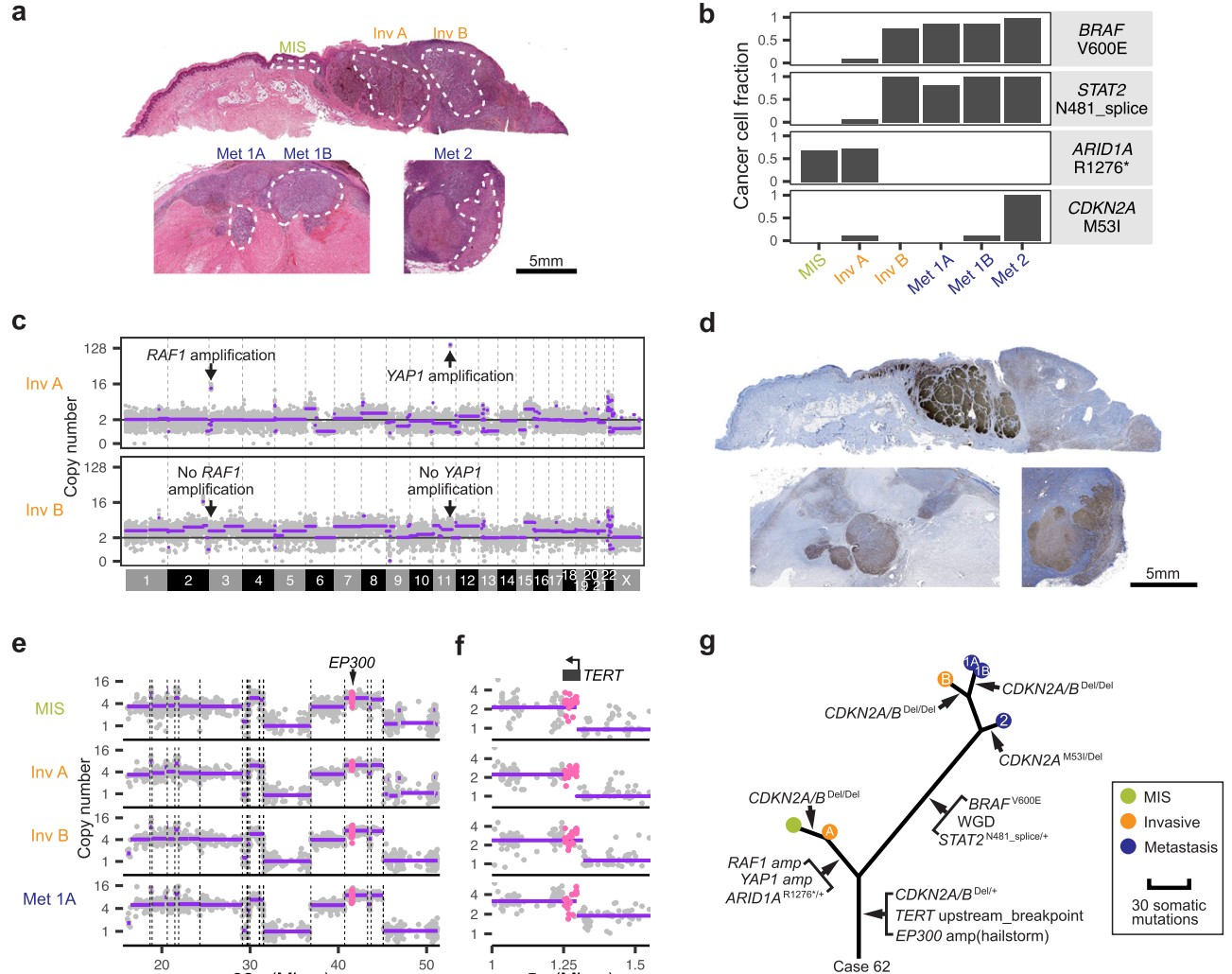

**Fig. 4 | Intratumoral heterogeneity of genetic alterations exemplified by case 62. a** Hematoxylin and eosin-stained sections of the primary tumor on the thumb and corresponding lymph node metastases. The in situ, and invasive areas of the primary and three areas of the corresponding metastases were microdissected as indicated by the dashed white lines. Scale bar: 5 mm. **b** The cancer cell fraction (CCF) of pathogenic or likely pathogenic mutations shows notable differences among different tumor areas. **c** The invasive area A (Inv A) that lacks the *BRAF* V600E mutation shows amplification of *RAF1*, whereas invasive area B (Inv B) shows amplification of *YAP1*, which can be traced back to the melanoma in situ (MIS) area

(not shown). **d** The immunohistochemistry for YAP1 visualizes the striking heterogeneity within the primary tumor. One tissue section from each of the primary tumor and the metastases was used. Scale bar: 5 mm. Complex copy number alterations on chromosome 22 (**e**) and structural rearrangement immediately upstream of *TERT* (**f**) are shared across all progression stages of the tumor with congruent copy number transitions. **g** The relevant genetic alterations shown in an inferred phylogenetic tree. See Figshare[17] for more detailed analyses. Source data are provided as a Source Data file.

A and the flanking melanoma in situ visualizes these two heterogeneous subpopulations (Fig. 4d). Whole-genome sequencing data of this case suggests that the amplicons of *RAF1* and *YAP1* likely represent circular extrachromosomal DNAs (ecDNA) (Supplementary Fig. 7). Despite these differences, all six tumor areas shared copy number changes including a hailstorm on chromosome 22 with congruent CNTs (Fig. 4e), a CNT upstream of *TERT* (Fig. 4f) and focal deletion of *CDKN2A/B*, supporting their common clonal ancestry (Fig. 4g). Additional details of case 62, including multiple independent genetic alterations inactivating *CDKN2A/B*, and full summary of all other cases can be found on Figshare[17].

Similar patterns of heterogeneous MAP-kinase pathway driver mutations were seen in other melanomas (Fig. 5a). Case 49 had an *NRAS* Q61L mutation exclusive to one area of the primary with an *ERBB2* amplification exclusive to the adjacent area. A metastasis derived from the latter area carried a *MET* amplification and a *GNAS* R201H mutation in addition to the *ERBB2* amplification. Case 61 had an *NRAS* G12C mutation

in the invasive portion and a metastasis, whereas the melanoma in situ instead had separate amplifications involving *KIT* and *CRKL*, both being known driver mutations of acral melanoma[28,29]. The pattern of non-overlapping MAP-kinase pathway drivers indicates that these mutations arose after tumor initiation. In cases 100 and A19T, *NF1* inactivation emerged only later during progression (Fig. 5b). Both cases had amplifications on chromosome 11q spanning the *CCND1*, *PAK1*, and *GAB2* genes, likely representing MAP-kinase pathway alterations preceding inactivation of *NF1*. Interestingly, the two metastases in case 100 each harbored different *NF1* mutations, indicating that they arose from independent events after separation from their shared ancestral clone. In cases 110 and 80, the homozygous deletion of *CIC*, a negative regulator of ERK signaling[30], and *PTPN11* R498Q were present only in the invasive areas and were absent from the in situ areas (Fig. 5c, d). The two invasive areas in case 110 further diverged, with each area acquiring different additional driver mutations. In cases 48 and 50, *KIT* mutations emerged after bi-allelic

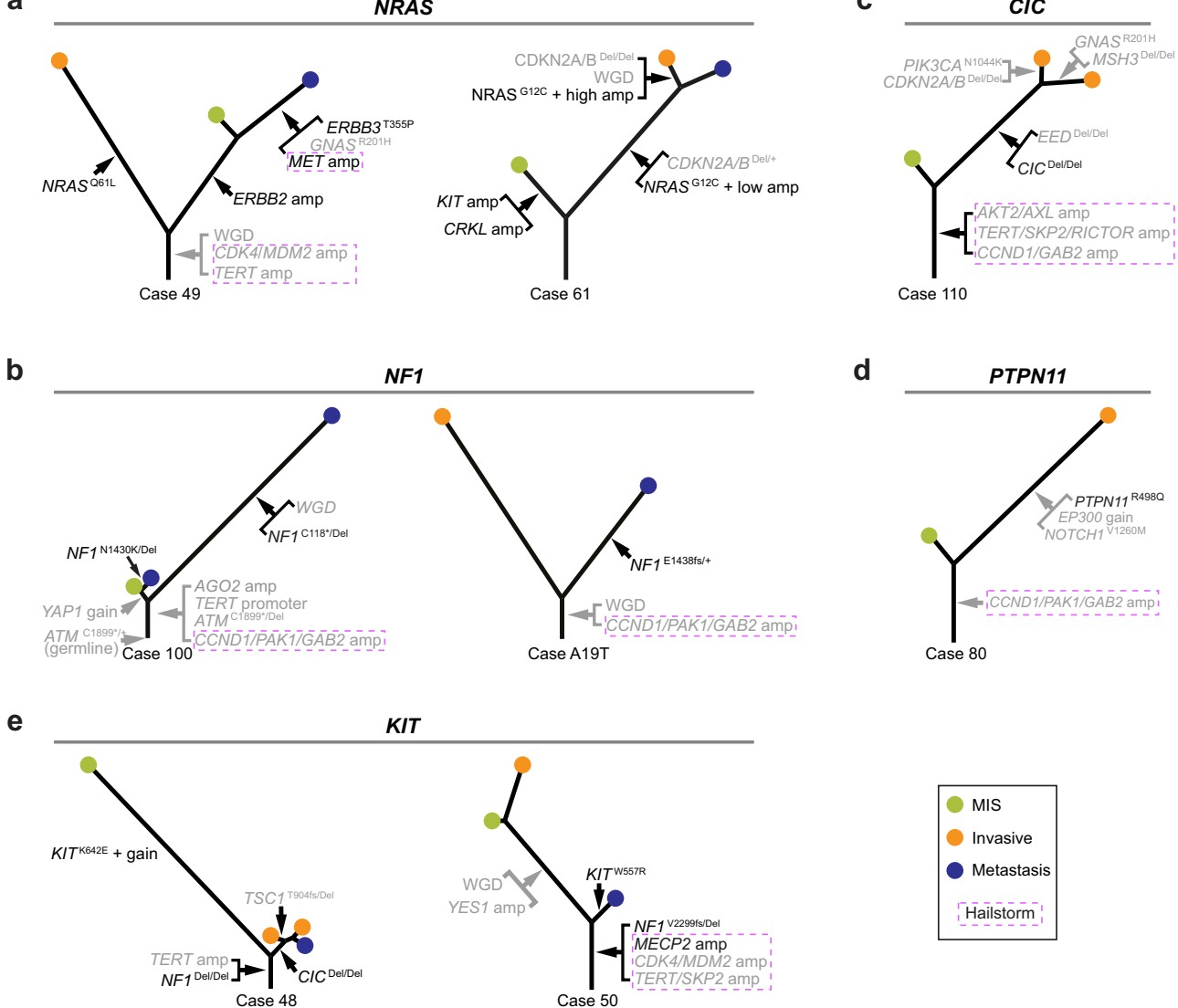

**Fig. 5 | MAP-kinase pathway driver mutations tend to arise after tumor initiation.** *NRAS* mutations (**a**), *NF1* mutations (**b**), *CIC* homozygous deletion (**c**), *PTPN11* mutation (**d**), and KIT mutations (**e**) arose after tumor initiation in multiple acral melanomas, placing them on the respective branches of the corresponding phylogenetic trees.

inactivation of *NF1* (Fig. 5e). In case 48, the *KIT*[K642E] mutation was only detectable in the in situ melanoma but absent from the invasive portion and metastasis, which instead showed homozygous deletion of *CIC*. In case 50, *KIT*[W557R] was observed only in the metastasis. The finding of multiple independent hits predicted to activate the MAP-kinase pathway is similar to what has been observed during the evolution of melanomas on the sun-exposed skin[12].

Genetic alterations disrupting other pathways important for melanoma development also arose later during progression. These include inactivating mutations or homozygous deletions of genes in G1/S cell-cycle checkpoint regulation (*CDKN2A/B*), the PI3-kinase (*PTEN*, *PIK3CA*, *TSC1*) and Hippo pathways (*MOB3B*) (Supplementary Figs. 2, 8), which were primarily encountered at the invasive or metastatic stages, albeit some genes in these pathways (*CDK4*, *CCND1*, *SKP2*, *RICTOR* and *YAP1*) were amplified early as part of hailstorms (Supplementary Figs. 2, 8). Point mutations of *CTNNB1* and *AXIN1*, likely activating the WNT pathway, also were restricted to invasive or metastatic stages (Supplementary Fig. 8) often contributing to genetically heterogeneous subclones within the primary tumor.

## Metastatic clones may genetically diverge early during tumor evolution

The different patterns of genetic alterations between the primary melanomas and their metastases provided insight into the timing of metastatic dissemination in some melanomas. A case in point is melanoma A19T (Fig. 6a), in which the primary tumor sample and the metastasis shared only few copy number changes and somatic mutations, suggesting that the metastatic population diverged early during the evolution of the primary. Similar patterns were apparent in cases 49 and 50 (Fig. 6a). Interestingly, in case 49, the metastasis was phylogenetically closer to the in situ portion of the primary than to its invasive component, indicating tumor cells may seed metastases early during the evolution of the primary tumor.

In case 102 (Fig. 6b), a metastasis in the brain was phylogenetically closer to one of two lymph node metastases than to the primary areas or the other lymph node metastasis, suggesting that this metastatic lineage branched earlier from the primary. In case 100 (Fig. 6c), the brain metastasis had a highly increased mutation burden with over 300 private mutations, likely attributable to temozolomide (SBS11) (Supplementary Fig. 6). It branches earlier from the primary than a

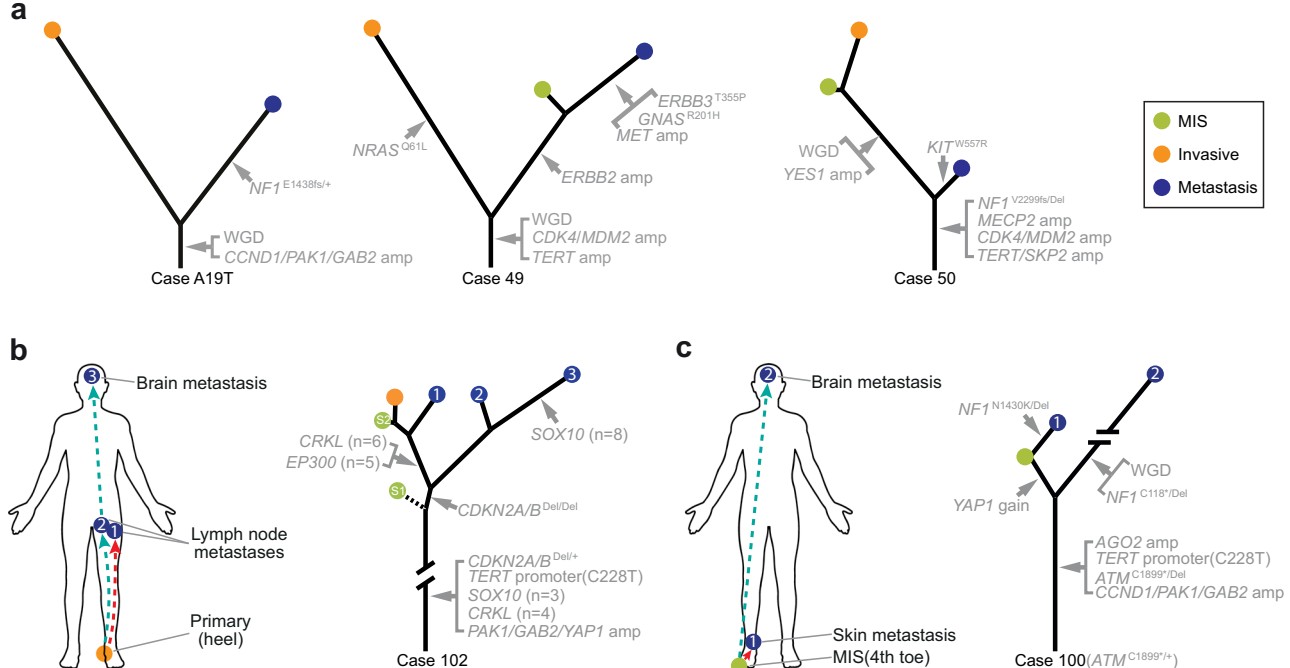

**Fig. 6 | Genetic differences of metastases and primary melanoma indicate early dissemination of metastatic clones. a** Evolutionary trees for three cases, one with a cutaneous metastasis and two with lymph node metastases showing comparatively early branching of the metastatic lineages from the corresponding primary melanomas. In two other melanomas multiple metastases were analyzed (**b**, **c**). The lineages of the brain metastases diverge early from the primary in both patients. The in situ portion of case 102 was computationally inferred to contain two different subclones. **b**, **c** Created with BioRender.com released under a Creative Commons Attribution-NonCommercial-NoDerivs 4.0 International license (https://creativecommons.org/licenses/by-nc-nd/4.0/deed.en).

cutaneous metastasis near the primary, also suggesting early divergence of the brain metastasis.

## Discussion

Our study reveals that the characteristic genomic alterations of acral melanoma, clustered CNTs with high-level amplification of a recurring set of oncogenes, arise before their earliest clinically identifiable progression stage, melanoma in situ (Fig. 7). The hailstorms in acral melanoma have features of what has been referred to as chromothripsis with abundant high-level amplicons[31] and tyfonas[9].

New hailstorms only rarely arose once the melanomas progressed to invasive and metastatic stages. The early and punctuated pattern in which hailstorms arose implicates a transient phase of genomic instability that abates later during progression (Fig. 7). Telomere crisis is a possible candidate mechanism of this pattern. When telomeres become critically short, chromatids can fuse and form dicentric chromosomes, inducing breakage-fusion-bridge cycles, which in turn can trigger chromothripsis[22,23,32]. Our observation that genomic alterations of *TERT* are a common and early event in acral melanoma would support the notion that hailstorms result from telomere crisis as they ceased to form at later progression stages at which telomerase was already genetically activated. We did not identify mutations in other genes involved in telomere maintenance in the minority of acral melanoma without *TERT* alterations but these may have cryptic alterations[24] or activation of the ALT (alternative lengthening of telomeres) pathway[33]. Once telomere maintenance mechanisms become activated, they are expected to stabilize telomeres and reduce the likelihood of telomere fusions possibly explaining the cessation of hailstorm formation. Another conceivable mechanism that could underlie the punctuated emergence of hailstorms involved intrachromosomal breaks outside of telomeres. Acral melanomas more frequently arise at anatomic site exposed to physical pressure or a history of trauma[20,34]. Physical force can result in rupture of the nuclear lamina, which can cause genomic damage including double-stranded DNA

breaks[21]. If it were physical force-induced DNA breaks rather than eroded telomeres that kick off the cascade leading to hailstorms, the early selection for telomerase activation would likely result from telomere-mediated chromosomal healing. This process involves capping double-stranded DNA breaks with new telomeres and would be advantageous to cells as it would prevent continuous genomic reshuffling and losing any genomic constellation that increased cellular fitness. A recent comparative analysis of whole genome sequencing data from various melanoma subtypes has found longer telomeres in acral melanomas compared to cutaneous melanomas[35]. This finding could be supportive of the second mechanism. In cutaneous melanoma the selective advantage for *TERT* promoter mutations arises from critically short telomeres, and mutational activation of *TERT* expression does not result in a net re-extension of telomeres, leaving telomeres short[36]. Longer telomeres in acral melanoma could indicate that *TERT* activation was not selected by critically short telomeres but to re-cap the ends of broken chromosomes of cells that had not yet exhausted their telomeres.

The distribution of hailstorms across the genome was nonrandom and frequently involved genomic regions with oncogenes, such as *CCND1*, *PAK1*, *GAB2*, *CRKL*, *TERT*, and *CDK4*. Each of these genes has functionally been implicated in melanoma progression, suggesting that hailstorms are under positive selection and actively drive tumor progression. In situ studies of primary acral melanomas have traced the amplifications of genes, such as *CCND1* and *TERT* through different progression stage of the primary tumor[37,38]. When amplification is observed in the invasive portions of acral melanoma, in situ hybridization reveals that cells in the adjacent in situ portion, and even to the morphologically normal-appearing melanocytes in the flanking epidermis also harbor the amplification. These 'field cells' sometimes extended 20 mm or more into what clinically and histopathologically appeared like normal skin[37,38]. Acral melanomas are notorious for recurring at the excision site of the primary melanoma, when not excised with wide safety margins[39], and field cells likely

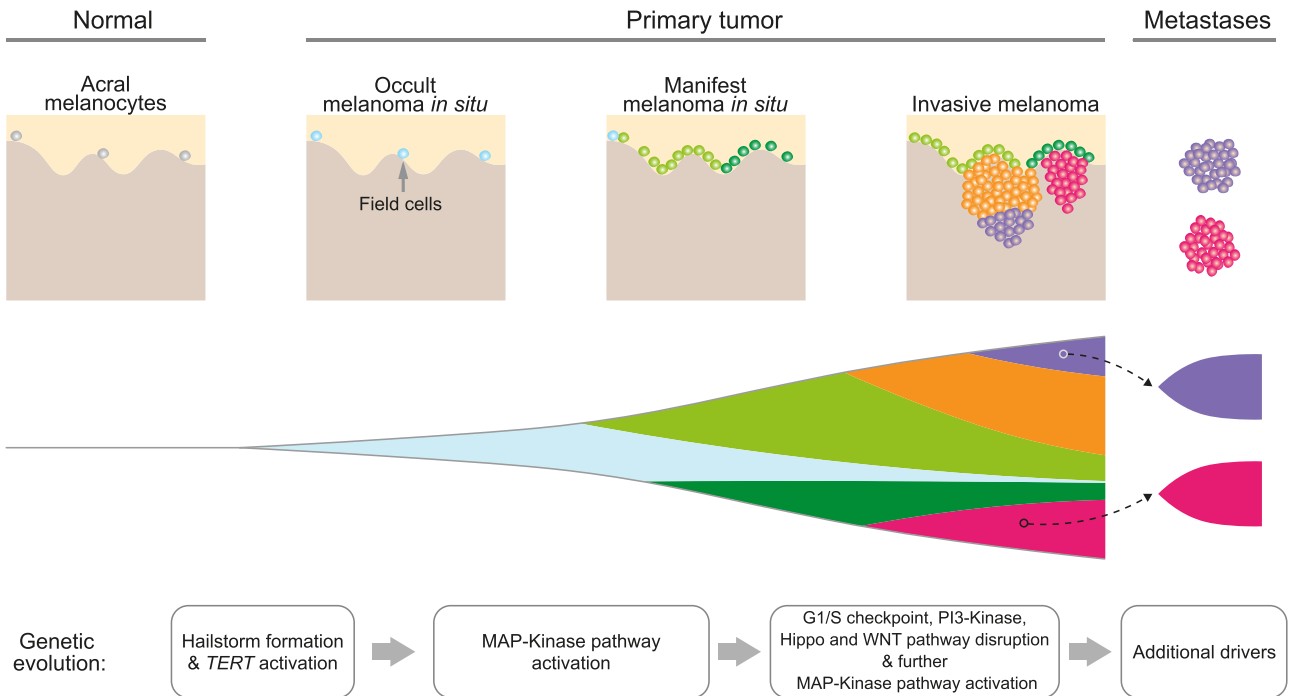

**Fig. 7 | A model of the genomic evolution of acral melanoma.** The progression cascade begins with field cells, i.e., histopathologically normal acral melanocytes distributed along the basilar epidermis at normal density but already harboring hailstorms as demonstrated by prior FISH studies[37,38]. As cell density increases, the nascent tumor becomes histopathological and clinically manifest as melanoma in situ, which later develops foci of dermal invasion with genetically heterogeneous subclones that can seed metastases. Activation of *TERT* arises before canonical driver mutations of melanoma, which activate the MAP-kinase pathway, override the G1/S checkpoint, and further disrupt other pathways.

represent occult disease that leads to local recurrence after apparently complete excision.

We observed a UV-radiation-induced mutational signature in a subset of acral melanomas. The distribution of UV signature mutations was heterogeneous across different tumor areas, indicating that UV radiation operates on partially transformed cells rather than normal melanocytes. These cells may expand from sun-shielded into sun-exposed regions and thereafter receive UV radiation. Notably, cells of overt melanoma in situ can reside in the most superficial skin, whereas normal acral melanocytes are situated in the basal layer of the epidermis, often beyond the reach of UV radiation in body sites with thick skin.

The phenomenon of field cells, which are only detectable because of their amplifications, confirms that hailstorms are an early, and possibly an initiating event in the evolution of acral melanoma. By contrast, activating mutations of the MAP-kinase pathway, which constitute initiating drivers of cutaneous and uveal melanomas, appear to arise later during the progression of acral melanoma (Fig. 7). This indicates that hailstorms likely are sufficient to drive the initial clonal expansion but require cooperating driver mutations to complete the evolutionary trajectories of acral melanomas to form invasive and metastatic cancers. This deferred selection for MAP-kinase pathway mutations results in an unexpected heterogeneity in the primary tumors, with *BRAF* V600E mutation or *NRAS* Q61 mutations present in some areas of the tumor and absent in others, a finding that is highly unusual for cutaneous melanomas.

Phylogenetic analyses of primaries and corresponding metastases revealed that some metastases were phylogenetically closer to the in situ than the invasive component and indicating that metastatic dissemination can occur comparatively early during tumor evolution.

The differences between acral and cutaneous melanomas may reflect differences in the cell of origin of these tumors. Recent studies have identified differences between the transcriptional programs of volar melanocytes, the likely cell of origin of acral melanoma, and

melanocytes from the hair-bearing skin[40]. The *CRKL* gene, which was recurrently amplified in our and other studies of acral melanoma, has an oncogenic role specifically in volar melanocytes[29]. Despite the remarkable differences between acral and non-acral melanomas, their evolutionary trajectories converge during later progression stages, with incremental accrual of additional MAPK pathway mutations, inactivation of *CDKN2A/B* at the transition to the invasive stage, and the inactivation of *PTEN* in invasive and metastatic stages[10,12] (Fig. 7). Our cohort was too small to detect any differences in the genetic alterations across different acral sites such as nail apparatus, palm or sole. This limitation also extends to the comparison of acral melanomas from various world populations. The distribution of hailstorms and their emergence early during evolution was consistent across these sites and populations included in our study.

In summary, our study reveals distinctive features in the genetic evolution of acral melanoma, a subtype that arises at anatomic sites with little or no exposure to UV radiation. Our results implicate complex aberrations as the initiating events of acral melanoma, with the underlying double-stranded DNA break, caused by telomere crisis or physical force, as a key mutational mechanism in the activation of oncogenes. The observation of *TERT* alterations as the earliest driver mutation and the punctuated pattern in which hailstorms arise points at telomerase as a critical factor in ending this transient phase of genomic instability. This transient phase during tumor evolution likely generates fields of pre-neoplastic cells from which heterogeneous subpopulations of cancer cells emerge that are driven by diverse gain of function alterations. These evolutionary features may explain the aggressive clinical course and relative resistance to targeted and immune-based therapy of these melanomas.

## Methods
### Description of cohort
This study was approved by the Institutional Review Board at the University of California, San Francisco (Parnassus Committee), and

regional Ethical Committees at Kumamoto University (the Ethics Committee for Human Genome and Gene Analysis Research, Graduate School of Life Sciences), National Taiwan University Hospital (Research Ethics Committee C), University of Zurich (Kantonale Ethikkommission Zurich), and Hospital Obrero (Research Ethics Committee of Hospital Obrero, La Paz). Our cohort includes acral melanomas from 37 patients, 2 of which were previously reported as case E from ref. [15] and case TCGA-ER-A19T from the TCGA-SKCM project[16]. The formalin-fixed paraffin-embedded tissues of 35 patients were retrieved from the archives from Taiwan, Japan, Switzerland, and Bolivia. The respective Institutional Review Boards waived the necessity for patient consent, considering that this is a retrospective analysis of left-over archival tissue specimens. For each patient we included two or more tumor areas of different tumor stages (in situ, invasive areas of the primary tumor or tissue from metastases). Non-lesional tissues were used as a source for normal DNA and available for all except one patient.

## DNA extraction and exome sequencing

For each patient, images of 5 μm thick hematoxylin and eosin-stained sections of FFPE blocks were reviewed by two pathologists to identify tumor and adjacent non-lesional areas. Microdissection of these areas was performed on 10 μm sections under a dissection microscope. Genomic DNA was extracted using AllPrep DNA/RNA Micro Kit (80234, Qiagen) or DNAstorm FFPE extraction Kit (CD503, Cell Data Sciences) and quantified using the Qubit 2.0 Fluorometer with the dsDNA High Sensitivity assay Kit (Q32854, Thermo Fisher Scientific). For each sample, 50–333 ng of genomic DNA was sheared to ~200 bp using a Covaris E220 Sonicator. Libraries were generated using the KAPA HyperPrep Kit (KK8505, Roche) and quantified by qPCR using the KAPA library quantification Kit (KK4854, Roche). The library size distribution was determined using an Agilent 2100 Bioanalyzer with the High Sensitivity DNA Kit (5067–4626, Agilent). We next used the SureSelect XT HS reagent Kits (5190–9685 and 5191–6686, Agilent) and the SureSelect XT HS Human All Exon V7 target-enrichment probes (5191–4028, Agilent) for exome capture. The captured libraries were sequenced on an Illumina NovaSeq 6000 instrument with 150 bp paired-end reads.

## Processing of sequencing data

We mapped FASTQ data to the hg19 genome assembly using the Burrows-Wheeler Aligner[41]. Picard was used to remove PCR duplicates and calculate insert size and sequencing coverage of exonic regions, followed by GATK to perform base quality score recalibration and realign indel regions.

## Verifying the genetic ancestry of patients

To verify the genetic ancestry of included patients from our cohort, we downloaded the phase 3 autosomal variants of the 1000 Genomes Project (ftp://ftp.1000genomes.ebi.ac.uk/vol1/ftp/release/20130502; PED file from: ftp://ftp.1000genomes.ebi.ac.uk/vol1/ftp/technical/working/20130606_sample_info/20130606_g1k.ped) and removed multi-allelic calls, indels and SNPs with allele frequency <1%. The genomic coordinates of the remaining SNPs were obtained and the corresponding genotypes of non-lesional normal samples from our cohort were determined by using SAMtools and BCFtools. We labeled loci with fewer than 10 reads as having unknown genotype and excluded those from subsequent analyses if half or more of the patients showed unknown genotype. Loci with alternative nucleotides not represented in the 1000 Genomes Project were also removed. We next applied PLINK[42] v1.90b6.24 for downstream analyses. Specifically, data from both our own cohort and the 1000 Genomes Project were converted to PLINK format and merged. From the merged dataset we removed SNPs with allele frequency <10% and further pruned to obtain the subset in approximate linkage equilibrium with arguments "--maf 0.1 --indep 50 5 1.5". The final set of 28557 SNPs were used for PCA analysis.

## Calling somatic mutations

We used Mutect2[43], Strelka2[44], and FreeBayes[45] to call somatic mutations and indels in each tumor sample against its corresponding normal. For Mutect2, a panel of normal (PON) reference was used, which was assembled from all germline samples of the cohort. For each patient, all candidate mutations that were labeled as "PASS" by at least two of the three tools in any sample were merged and annotated using ANNOVAR[46]. Common SNPs with a population frequency greater than 1% in any of the 5 databases (ExAC, GnomAD, 1000 Genome, dbSNP150, and NHLBI-ESP) were removed. We further manually inspected indels and mutations that are flagged as "clustered_events" or "clustered_events;haplotype" by Mutect2 using Integrated Genome Browser (IGV). For the filtered mutations, we used the "samtools mpileup" command to re-count reads from the mutated and reference alleles. This helps the accurate calculation of mutant allele frequency by correcting for overlapping read pairs, as well as enables the detection of mutations in samples whose mutant allele frequencies were too low, due to either low clonality or low tumor purity. Mutations that were supported by less than 4 mutated reads in all samples of a patient were additionally excluded. All somatic mutations are listed in Supplementary Data 2.

## Calling germline mutations

We used HaplotypeCaller to call germline mutations in the normal DNA data for each patient. Mutations with fewer than 4 reads or an allele frequency under 25% were removed and the remaining mutations were annotated using Annovar. We further removed common SNPs with >1% mutation frequency in any of the 5 databases (ExAC, GnomAD, 1000 Genome, dbSNP150, and NHLBI-ESP). The pathogenicity of germline mutations was estimated using annotations from OncoKB and ClinVar. Mutations that were annotated as "benign/likely_benign" by Clinvar, as "neutral" or "inconclusive" by OncoKB, or were located very close to protein C-terminus were not considered as likely driver events. Overall, two heterozygous germline mutations, *MITF* [E381K] (case 58) and *ATM* [C1899*] (case 100), were identified as being probably pathogenic.

## Allelic copy number analysis and estimation of tumor purity

We used CNVkit[47] v0.9.6 to calculate the bin-level $\log_2$ ratio (logR) of target and anti-target regions of each tumor sample, with a pooled reference created from normal samples of all female patients. The two previously published cases were analyzed separately, considering their potential difference in target regions. The bins were segmented by using the default CBS (circular binary segmentation) method. After segmentation, we used the "cnvkit.py call" command with the "-m clonal" argument to estimate the absolute copy number of the major and minor alleles of each segment. For the "--purity", "--center-at" and "--vcf" arguments, we provided tumor purity, the average logR of clonal diploid segments (see below), and the VCF output of FreeBayes which contained B-allele frequency of germline variants. All copy number calls are listed in Supplementary Data 3.

We used several approaches to estimate tumor purity. First, we employed the shift of allele frequency of germline SNPs at regions with copy loss LOH (loss of heterogeneity), calculated similarly as previously reported[12]. Supposing the median allele frequency of the minor allele in tumor is $A$, then tumor purity can be estimated as:

$$2 - \frac{1}{1-A} \tag{1}$$

Second, the shift of allele frequency of germline SNPs can also be applied to regions with copy-neutral LOH. Supposing the median allele

frequency of the minor allele in tumor is $A$, then tumor purity can be estimated as:

$$1 - 2A \qquad (2)$$

Third, from the segmented CNA data from CNVkit, we obtained the median logR values of haploid and diploid segments that were longer than 5 megabases and not from chromosome X of male patients and chromosome Y. To ensure the accurate identification and avoid subclonal CNA segments, we inspected the distribution of logR values, the BAF plots, and further cross-referenced the solution from FACETS[48], another tool that estimates tumor CNAs. Tumor purity and ploidy were resolved by using the following formula[49]:

$$\log R = \log_2 \left( \frac{2 * (1 - \rho) + N * \rho}{2 * (1 - \rho) + \varphi * \rho} \right) \qquad (3)$$

Here $\rho$ and $\varphi$ represent tumor purity and ploidy, respectively. The logR of haploid and diploid segments ($N = 1$ and 2, respectively) were identified above. In samples with no or very limited haploid segments, we used the logR of triploid segments together with diploid regions, to resolve tumor purity and ploidy using the same formula.

When possible, both germline SNP (either approach 1 or 2) and CNA segmentation (approach 3) approaches were applied for a sample. The estimates were mostly very close and thus averaged for downstream analyses. Meanwhile, if a sample had very limited CNA or low purity that rendered the above approaches unapplicable, we applied the MAF (mutant allele frequency) of clonal heterozygous somatic mutations at diploid regions. Specifically, supposing the median MAF is $A$, then tumor purity is estimated as $2*A$.

We considered a sample underwent WGD if over 50% of its autosomal genome has a major allele copy number ($N_{major}$) of at least 2, similar as previously defined[50]. The major allele of a CNA segment is the parental allele with higher copy number. All samples were further manually inspected. For the samples with WGD, we found that on average 89% (all >68%) of the genome showed $N_{major} \geq 2$.

### Calculation of cancer cell fraction (CCF) of somatic mutations
For each somatic mutation, CCF was calculated separately for each sample of an acral melanoma, using the following formula[51]:

$$CCF = \frac{f}{m * \rho} (\rho * N + (1 - \rho) * n) \qquad (4)$$

Here $f$, $\rho$, $N$, and $n$ denote MAF, tumor purity, and the total absolute copy number of a locus in tumor and normal, respectively. Moreover, $m$ represents the multiplicity of the somatic mutation, indicating the number of DNA copies that harbor the mutation, and is a positive integer at regions with clonal CNA. For instance, at regions with copy neutral ($N = N_{major} = 2$) LOH, m can be either 1 (mutated after the duplication of the retained copy) or 2 (mutated before the duplication). We calculated the likelihood of $f$ (i.e., MAF) supposing 1 … $N_{major}$ copies being mutated with a binomial distribution. The value with the highest likelihood was assigned to $m$. The 95% confidence interval of CCF was estimated by using Bootstrap to resample the number of references and alternative reads for 10,000 times.

### Identifying hailstorm aberrations based on copy number data
We observed pervasive high-level amplifications in our cohort, the majority of which fell into genomic regions that also contained multiple low-level copy number alterations, resulting in complex patterns of copy number alterations with high densities of CNTs. Visual inspection of our cohort showed that these aberrations were frequently shared across all tumor areas of a patient, suggesting that they probably represent "one-off" genomic events that were generated

within one or few cell cycles early during progression. To formally identify such hailstorm aberrations, we first defined CNTs as breakpoints that separated neighboring segments by a logR of 0.5 or more, adjusted for purity and ploidy. We defined chromosomal arms with hailstorms using the following criteria:

1. Increased CNT density compared to genome average. This was determined based on a hypergeometric test, using the number of CNTs of a specific chromosomal arm, the total number of CNTs across the genome, the number of base pairs of the specific chromosomal arm, and that of the rest of the genome as inputs. A cutoff of $P < 0.0001$ was applied.
2. At least one segment amplified to more than threefold above background ploidy and a minimal size of 200 kb.

Sometimes CNTs were clustered only in part of a chromosomal arm, which rendered the arm-based CNT density test less sensitive. To address this, a sliding window approach was additionally performed for chromosomal arms of sizes above 50 Mbps. Specifically, for each sliding window of 50 Mbps with step size of 5 Mbps, the same tests described above were applied.

In addition, the boundaries, i.e., the left and right most CNTs, of the aberration were determined. If the overall width was narrower than 5 Mb, then the aberration was not considered.

Chromosomal arms and genomic regions that passed the above filters were considered as harboring hailstorm aberrations. Due to variation in tumor purity and the performance of copy number segmentation, a hailstorm aberration shared by all tumor areas of a patient may not pass filtering in some of them. To address this, if any tumor sample of a patient passed the above filters, we manually inspected all other tumor samples of same patient to determine whether the hailstorm was shared among different samples.

### Estimating the frequency of hailstorms in cutaneous melanomas
We downloaded the whole exome sequencing data of melanomas from the TCGA-SKCM project[16], which were already mapped onto the hg38 genome assembly and in BAM format. We first filtered the dataset by removing: (1) known acral or mucosal melanomas based on the pathology reports on cBioPortal[52], (2) tumors with no or low UV exposure, i.e., <50% of SNVs being [C/T]C > T and less than 2 CC > TT dinucleotide substitutions, and (3) tumors with purity smaller than 30%. For the remaining 358 melanomas, we then applied the same statistical approach as for acral melanomas to identify hailstorms. Copy number outputs from FACETS were used for this analysis.

We note that it is conceivable that a few acral or mucosal melanomas that arose from sun-exposed sites may still be accidently included by this approach since the primary sites of most of the metastases and some of the primary melanomas were not specified.

### Whole genome sequencing of samples from cases 62 and 110
We conducted whole-genome sequencing for cases 62 and 110, including the normal, in situ, and two invasive areas for each case, resulting in a total of eight samples. The sequencing reads were aligned and processed in the same manner as those for exome sequencing data. The average deduplicated sequencing depth was 53X (45-61X). We applied Delly2[53] to identify SVs and used AmpliconArchitect[54] to detect probable ecDNAs.

### Construction of phylogenetic trees
Phylogenetic trees were constructed manually for each case using the principle of maximum parsimony. The trunk length reflects the number of common mutations, while internal and terminal branch lengths indicate mutations private to multiple areas and a specific area, respectively. Only somatic mutations were included. All phylogenetic trees were rooted to the germline state, with mutations with CCF $\geq 0.5$ considered for each area.

## Signature analysis of somatic mutations

For signature analysis, we used deconstructSigs[55] to decompose somatic mutations into COSMIC v2 signatures. For each patient's melanoma, signature analysis was performed for the trunk and branches, if they had a minimum of 35 SNVs, with few exceptions. From the output, we focused on a few prevailing signatures, including SBS1 (aging), SBS2/13 (APOBEC), SBS3 (defective homologous recombination DNA damage repair), SBS7 (UV exposure) and SBS11 (temozolomide treatment).

## Targeted sequencing of a panel of cancer genes

For validation purpose, we further performed deep sequencing on a customized small panel of 80 cancer genes in 70 tumor samples. The panel was manufactured by Integrated DNA Technologies and was designed to capture exons as well as certain noncoding regions such as *TERT* promoter. The bait region can be seen in Supplementary Data 4. The post-captured libraries were sequenced on an Illumina HiSeq 4000 instrument with 100 bp paired-end reads.

## Immunohistochemistry (IHC) and fluorescence in situ hybridization (FISH)

IHC was performed with the following commercial antibodies: YAP1 (12395S, Cell Signaling Technology) and p16 (BSB-5828, Bio SB). FISH was conducted as previously described[56] with minor modification. Briefly, 5 μm sections were mounted onto charged slides and baked overnight at 56 °C. The slides were then treated to remove paraffin, submerged in 1×saline sodium citrate (SSC) pH 6.3 at 80 °C for 35 min and 2×SSC for 2 min, and washed. After protease digestion (0.5 mg pepsin/ml in 0.01 N HCl) at 37 °C for 20 min, sections were washed, dehydrated, and dried. Hybridizations were carried out at 37 °C for 16 to 18 h. Following hybridization, sections were washed (2×SSC/0.3% NP40) at 73 °C for 2 min, then at room temperature for 1 min, and air dried. DAPI (4′, 6-diamidino-2-phenylindole) (06J49-001, Abbott) was then applied. *CDKN2A* and control probes for FISH was from Biocare Medical (PFR7008A, https://biocare.net/product/cdkn2a-p16-9p21-copy-control-9/).

## Reporting summary

Further information on research design is available in the Nature Portfolio Reporting Summary linked to this article.

## Data availability

The BAM format raw whole-exome, whole-genome, and targeted sequencing data of all samples generated in this study have been deposited to the dbGaP database under accession number phs003451. The sequencing data are available under restricted access for privacy consideration and regulatory compliance. Permanent employees of an institution at a level equivalent to a tenure-track professor or senior scientist with laboratory administration and oversight responsibilities may request access through dbGAP. The sequencing data of melanomas from the TCGA-SKCM project were downloaded from the NCI Genomics Data Commons, under dbGaP accession number phs000178. The sequencing data of case E is available from dbGaP under accession number phs000941.v1.p1. Summary of genetic findings of all 37 cases included in this study can be found at Figshare [https://doi.org/10.6084/M9.FIGSHARE.22773791][17]. Source data are provided with this paper.

## Code availability

Data analyses was conducted using publicly available software packages. The code used for generating figures in the article can be accessed at: https://github.com/Bioinfowangm/Genetic-evolution-acral-melanoma.

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

## Acknowledgements

We thank all the included patients from which we obtained specimens and sequencing data. This study was supported by an NCI Outstanding Investigator Award (1R35CA220481, B.C.B), a Melanoma Research Alliance Team Science Award (B.C.B., I.Y. and A.H.S.) and Dermatology Fellows Award (ID:737988, https://doi.org/10.48050/pc.gr.141692; M.W.), a National Institutes of Health grant (R01AR080626; A.H.S.), and grants from the Ministry of Science and Technology of Taiwan (MOST 108-2314-B-002-176; Y.S.S.) and National Taiwan University Hospital (NTUH105-M3349 and NTUH106-M3737; Y.S.S.). Our analyses were performed on the C4 high-performance computing cluster in the Helen Diller Family Comprehensive Cancer Center. We thank Haruka Kuriyama from Kumamoto University for sample preparation, Raymond Cho for data sharing, Swapna Vemula, Jingly Weier, and Sonia Mirza for technical assistance, and members in the Bastian and Yeh labs for helpful discussion.

## Author contributions

M.W., B.C.B, A.H.S., I.Y. and. conceived the study. S.F., Y.S.S., E.R., M.S.A., M.L., R.D., J.Y.L. and C.Y.C. provided the clinical samples. I.Y. and B.C.B. performed tumor area selection. N.C.P., C.S. and S.L. prepared the sequencing libraries. M.W. performed bioinformatic analyses and data visualization. M.W., I.B., J.D.A., I.Y. and B.C.B. performed and interpreted in situ analyses. M.W., B.C.B., A.H.S., I.Y. and interpreted the genetic data and wrote the manuscript.

## Competing interests

The authors declare no competing interests.
