## [Peer Review File · Nature Communications]

REVIEWER COMMENTS

Reviewer #1 (Remarks to the Author): Expert in computational cancer genomics and evolution, genome instability, and complex structural variants

Wang, Bastian and colleagues report a very interesting whole exome profiling study of 139 acral melanoma samples from 37 patients, including in situ, primary, and metastatic lesions. The key finding is clustered amplifications in acral melanoma, which they call "hailstorms", appear to be both truncal and stable across melanoma evolution. In addition point mutations in MAPK pathway members, including those driven by UV mutations, occur later and are often not shared across clones. The study provides new contrasts between the evolution of acral and sun-exposed cutaneous melanoma.

Overall I think the findings are quite interesting. The key limitation is that this largely copy number (and fundamentally SV) story is studied using whole exome rather than whole genome sequencing. As a result the key claim that these events occur early and are stable can only be made approximately, since the copy number profiles from whole exome are noisy and approximate. While the several examples that they provide of early and stable amplifications are encouraging, they would be much more convincing if these were supported with whole genome data. In particular, am not sure the "hailstorm" name is useful since it is based on exome analysis, occurs very frequently (80%) and it is unclear how these patterns actually look on whole genomes.

Specific points

- The claim that hailstorms are truncal and stable is interesting, but rests on the comparison of patterns that by nature of whole exomes cannot be compared at breakpoint resolution to convincingly show this. Whole genome sequencing of tumors from a few cases with truncal amplifications cases would be useful to confirm that the SV breakpoints are shared with the in situ and that these are stable across samples. Alternatively, these patterns may be convergent evolution and may also be continuing to evolve, meaning accumulating additional breaks and gains. Which may be interesting, but a different result.
- The term hailstorm is interesting, but as noted by the authors other terms have already been applied to some of the complex copy number alterations in acral melanoma whole genomes, including amplified chromothripsis and tyfones. If hailstorms are just those same phenomena but projected into exome, do they need their own name? Alternatively they may be distinct, but it's not clear when and how. For example the EP300 amplification in Fig 4e does not seem to have that high of a copy number so maybe it's just a standard chromothripsis? The authors could for example create a virtual exome from previously published whole genome data to see how their criteria overlaps with previously described events. They could also profile some of their samples with whole genome sequencing (see above) to if and when these events are different.
- The hailstorms also seem very frequent (about 80% of acral melanomas). How frequently do they occur in other tumor types - such as sun exposed melanoma?
- Are some of the acral melanoma amplifications ecDNA? These may explain events that disappear or

fluctuate in copy number such as the YAP1 and RAF1 amplification in Figure 4c. FISH may be helpful to confirm.

- Do the APOBEC mutations co occur with amplifications? Are the APOBEC mutations also early? It may be useful to project COSMIC signatures on the phylogenetic trees. This is sort of done for UV in Figure 3 but does not appear to use COSMIC signatures (see below).

Minor points

- Criteria for hailstorms in the methods are unclear. How exactly is the hypergeometric test applied in each arm or 50 Mbp window to assess CNT density? Specifically, what data is put in the contingency table ?

- The text mentions various COSMIC signatures (such as SBS7) but then Figure 3 shows only UV mutations and defined differently (using criteria based on dinucleotides ie "[C/T]C>T or CC>TT"). Which is actually used in the analysis - COSMIC or this definition?

- deconstructSigs in line 535 is misspelled (deconstrucSigs)

Reviewer #2 (Remarks to the Author): Expert in acral melanoma genomics and intratumour heterogeneity

The authors use WES data from FFPE tissues of different countries to explore the genetic evaluation of acral melanoma. The quantity genetic analysis demonstrates the specific characteristics of different stage of acral melanoma. From the evolutionary trajectories of acral melanomas substantially diverge from those of melanomas on sun-exposed skin, where MAP-kinase pathway activation initiates the neoplastic cascade followed by immortalization later. The data demonstrate that punctuated formation of hailstorms, paired with early TERT activation, suggests a unique mutational mechanism underlying the origins of acral melanoma. There are several issue need make clearly.

1. Most of the cases come from East Asia. Does the racial disparities exist? How to avoid this? More and more Chinese acral melanoma patient data have been published and it should be included, at least using as a validation.

2. The results and conclusions need more validations at different levels. At genetic level, the WGS and EWS data from the Cell Discovery paper (Cancer Discov.2022 Dec 2;12(12):2856-2879.doi: 10.1158/2159-8290.CD-22-0603) need be cited and make combination analysis. So many RNA data could also be used.

3. Does the different specific anatomic position of acral melanomas (foot, finger, et al) affect the oncogenic events? Genetic data from Nature paper (Nature . 2022 Apr;604(7905):354-361. doi: 10.1038/s41586-022-04584-6.) need be analysed.

4. The genetic data get from the FFPE tissues. The results should be validated using fresh acral tissues. Even it is hard to get the TIS, invasion, and metastasis tissues from fresh samples, it could use the freeze section to make sure the quantity and quality of the different samples.

Reviewer #1 (Remarks to the Author): Expert in computational cancer genomics and evolution, genome instability, and complex structural variants

Wang, Bastian and colleagues report a very interesting whole exome profiling study of 139 acral melanoma samples from 37 patients, including in situ, primary, and metastatic lesions. The key finding is clustered amplifications in acral melanoma, which they call "hailstorms", appear to be both truncal and stable across melanoma evolution. In addition point mutations in MAPK pathway members, including those driven by UV mutations, occur later and are often not shared across clones. The study provides new contrasts between the evolution of acral and sun-exposed cutaneous melanoma.

Overall I think the findings are quite interesting. The key limitation is that this largely copy number (and fundamentally SV) story is studied using whole exome rather than whole genome sequencing. As a result the key claim that these events occur early and are stable can only be made approximately, since the copy number profiles from whole exome are noisy and approximate. While the several examples that they provide of early and stable amplifications are encouraging, they would be much more convincing if these were supported with whole genome data. In particular, am not sure the "hailstorm" name is useful since it is based on exome analysis, occurs very frequently (80%) and it is unclear how these patterns actually look on whole genomes.

We thank the reviewer for the comments. Before we address the concerns point by point, we would like to emphasize that the objective of our study was to determine the genomic evolution of acral melanoma. This scope includes the sequential order in which mutations in putative driver genes arise and the timing of the emergence of copy number changes. We deliberately chose whole exome sequencing for this purpose because it affords higher sequencing depth than whole genome sequencing. High coverage is critical to detect mutations in areas with low tumor cellularity such as melanoma in situ.

Specific points

- The claim that hailstorms are truncal and stable is interesting, but rests on the comparison of patterns that by nature of whole exomes cannot be compared at breakpoint resolution to convincingly show this. Whole genome sequencing of tumors from a few cases with truncal amplifications cases would be useful to confirm that the SV breakpoints are shared with the in situ and that these are stable across samples. Alternatively, these patterns may be convergent evolution and may also be continuing to evolve, meaning accumulating additional breaks and gains. Which may be interesting, but a different result.

We thank the reviewer for this comment, which allowed us to improve the manuscript and present additional data supporting our conclusions about the evolution of hailstorms. We agree that exome sequencing has limitation in mapping junctions of structural variations (SV) falling outside of exons at base pair level. However, we like to point out that the procedure of copy number analysis from whole genome and exome sequencing data is similar and involves splitting genomic regions into bins, calculating the GC-content corrected bin level log₂ ratios of read counts / sequencing depth between tumor and germline samples, and merging neighboring bins with similar values into larger segments. The bin level log₂ ratio values from exome sequencing are as accurate as those from whole-genome sequencing data. In fact, due to the higher coverage, exome sequencing provides less 'noisy' copy number data with increased discrimination of copy number transitions as it provides higher number of reads per bin. The difference in log₂ ratio values in bins flanking a copy number transition enables the identification of breakpoints at the

bin level, albeit not base-pair level. Despite this limitation SV junctions within hailstorms did occasionally fall within exons, allowing us to map junctions at base-pair resolution, which we now have added to the revised manuscript. To address the reviewer's point directly, we performed whole-genome sequencing for representative samples and identified shared SV junctions within hailstorms. The shared structural rearrangements at bin- and base pair level further establish that hailstorms are clonal across progression stages of individual acral melanomas.

In **Figures 1D** of the original manuscript, we displayed the shared hailstorms of case 110 showing congruent copy number patterns of the hailstorms on chromosomes 5p, 11q and 19q across all three tumor areas. In **Fig. R1** below we show this pattern at higher resolution with the vertical dashed lines assisting the viewer to note that the log₂ ratios of copy number transitions at the bin level are congruent across all three tumor areas of the same tumor. For the in situ portion, the copy number segmentation was expectedly more noisy, due to lower tumor cellularity of this area. We note that while it is possible if not unexpected that the hailstorm patterns may drift apart as subclones diverge as the evolution of the cancer proceeds, the numerous congruent copy number transition coordinates indicate that the underlying event occurred in a common ancestor.

Figure R1. Highly concordant bin-level log₂ Ratio values of hailstorms for the three tumor areas of case 110. The vertical lines represent the copy number transitions in invasive 1 and largely coincide with those from in situ and invasive 2.

In response to the reviewer's concern, we sought to further support this notion at base pair resolution. Although most SV breakpoints resided within noncoding regions not covered by the exome sequencing baits, some copy number transitions did occur in covered regions, permitting their precise mapping with whole exome sequencing data. For example, in case 110 we identified several SV junctions that were shared across all tumor areas. One such example is a 3' to 3' translocation between 5p and 19q (**Fig. R2A**, left), with the two ends of the junction being chr5:597,775 and chr19:49,298,816, respectively. In all three tumor areas, but not the adjacent normal, we detect junctional reads with identical breakpoints supporting this translocation. Another translocation between the two chromosome arms, with the same breakpoints at chr5:16792964 and chr19:35512545 (**Fig. R2A**, right). Across our cohort, we observed similar SV junctions within hailstorms which were shared across all tumor areas in 18 cases. This additional information further supports that hailstorms are clonal across different progression stages of individual tumors and is now included in the revised manuscript. Further examples of shared hailstorm SV junctions can be found in below **Fig. R2B**, and **Figure S5** of the revised manuscript.

Figure R2. IGV screenshots of exome-seq data showing shared hailstorm SV junctions for all tumor areas of case 110 (A) and case 49 (B). The dashed lines in each panel signify the breakpoints of SV junctions, with corresponding genomic coordinates indicated.

To directly address the reviewer's concern, we performed whole-genome sequencing of the normal, in situ, invasive 1 and invasive 2 areas of case 110. We processed the reads in the same manner as whole-exome sequencing data and applied Delly2 (Rausch et al. Bioinformatics. 2012) to detect SV junctions. The mean deduplicated sequencing depth was 50x (45-59x). For invasive 1 and invasive 2, which had high tumor purity, we identified 286 and 317 SV junctions respectively

across the genome. The majority of the SV junctions, 72.7% (208/286) and 72.2% (229/317), respectively, involved 5p, 11q and 19q (**Figure R3A**). Of these junctions, 150 were shared in both areas. The shared junction reads showed significantly higher variant allele frequencies (VAFs) than those that were private to either area (**Figure R3B**; $P \leq 0.001$, t-test), suggesting that the latter arose after the formation of hailstorms or are false positives. Interestingly, inter-chromosomal SVs only involved chr5 and chr19 indicating that they arose concurrently, whereas the hailstorm on chr11 likely arose independently. In the in situ area, we only detected 27 junctions on 5p, 11q or 19q, which we attribute to the lower sensitivity, due to the much lower tumor purity in situ lesions. 81.5% (22/27) of the SVs in the in situ were also present in invasive 1 and invasive 2, testifying to their clonal relationship. Using IGV to manually inspect SVs present in both invasive areas but not detected by the algorithm in the in situ area, we frequently identified junction reads supporting the presence of the SVs in the in situ (an example in **Figure R3C**). The copy number transition derived from whole-genome sequencing data also showed high concordance among the three tumor areas (**Figure R3D**), as seen in the exome sequencing data. This figure is incorporated as **Figure S4** in the revised manuscript.

Hailstorms recur at specific genomic regions as we highlight in the manuscript, indicating that they are under positive selection. While this renders convergent evolution a theoretical possibility, the highly concordant genomic patterns of hailstorms in different tumor regions within individual patients, but different patterns of hailstorms at similar regions from one patient to another argue against it. The finding of shared structural rearrangements supported by base level junctions and copy number transitions in whole genome sequencing and exome sequencing data provide very strong evidence of the clonal nature of hailstorms and render converging evolution a highly improbable mechanism. We have revised the manuscript to incorporate the data on clonal SVs derived whole-genome sequencing and re-analysis of the exome sequencing data (**Figures S4 and S5**).

Figure R3. Structural variation (SV) junctions and copy number profiles from whole-genome sequencing data. A) Circos plots showing SV junctions on chromosomes 5, 11 and 19. Inter- and intra-chromosomal junctions are colored gray and orange, respectively. The outer track shows the chromosome ideograms; the middle track shows the log₂ ratio of tumor to normal copy number (purple dots) with the dashed black line indicating 0. **B)** Variant allele frequencies (VAFs) of junctions shared between the two invasive areas were significantly higher than those unique to either area. Top: invasive 1; Bottom: invasive 2. **C)** An example SV initially undetected in *in situ* was, in fact, shared across all three tumor areas. **D)** Heatmaps based on whole-genome sequencing data supported highly concordant copy number profiles on 5p, 11q and 19q.

- The term hailstorm is interesting, but as noted by the authors other terms have already been applied to some of the complex copy number alterations in acral melanoma whole genomes, including amplified chromothripsis and tyfonas. If hailstorms are just those same phenomena but projected into exome, do they need their own name? Alternatively they may be distinct, but it's not clear when and how. For example the EP300 amplification in Fig 4e does not seem to have that high of a copy number so maybe it's just a standard chromothripsis? The authors could for example create a virtual exome from previously published whole genome data to see how their criteria overlaps with previously described events. They could also profile some of their samples with whole genome sequencing (see above) to if and when these events are different.

We deliberately chose a new term, because the existing terms amplified chromothripsis and tyfonas require whole genome sequencing data for their definition. It would thus be not formally correct for us to use these names because we used exome sequencing, for the reasons outlined above. We believe that for the case of acral melanoma, all three terms indeed capture the same biological phenomenon. We note that we originally described this phenomenon in acral melanoma more than 20 years ago (Bastian BC et al. Cancer Res 1998 and 2000, Curtin JA et al. N Engl J Med 2005). With the methods available at the time, the phenomenon manifested as copy number changes with specific features, namely focused high-level amplifications.

We originally used more descriptive terms in prior versions of the manuscript but ultimately decided that we needed a concise term that could be clearly defined with the data at hand to improve clarity and readability. We consider hailstorm a fitting term considering the wide adoption of kataegis (rainstorm) in the cancer genetics field.

- The hailstorms also seem very frequent (about 80% of acral melanomas). How frequently do they occur in other tumor types - such as sun exposed melanoma?

This question was answered in principle by our prior studies using comparative genomic hybridization (Bastian BC et al. Cancer Res 2000, Curtin JA et al. N Engl J Med 2005) which were carried out after we first noted this feature of acral melanoma (Bastian BC et al. Cancer Res 1998). These studies showed that the focused amplifications (associated with hailstorms) are significantly more common in acral and mucosal melanomas compared to melanomas on the sun-exposed skin. These findings were confirmed by subsequent next-generation sequencing studies (Hayward et al. Nature 2017)

Figure R4. An example of a cutaneous melanoma with hailstorms. A) Hematoxylin and eosin staining image of the tumor from cBioPortal. The scale bar is unknown. **B)** The copy number pattern of the hailstorms on chromosomes 5q and 22q.

To address the reviewer's question directly, specifically using our definition of hailstorms, we analyzed the whole exome sequencing data of melanomas from the TCGA-SKCM project. The raw sequencing data available for download were BAM format with reads already aligned to hg38 human genome assembly. From a total of 471 melanomas, we removed: 1) known acral/mucosal melanomas based on pathology report on cBioPortal, 2) tumors with no or limited evidence of UV exposure, i.e., <50% SNVs being [C/T]C>T and less than 2 CC>TT substitutions (as these melanomas from sun-shielded sites may be enriched for acral and mucosal melanoma), and 3) tumors with a cancer cell fraction lower than 30%. For the resulting 358 melanomas, we applied the same statistical approach as described in the manuscript and found 27 (7.5%) tumors with hailstorms. One such example is TCGA-BF-A5EQ which was a primary melanoma from trunk of a Caucasian patient diagnosed at the age of 63 (**Figure R4**). This suggests that the frequency of hailstorms in sun-exposed melanomas is around one-tenth of that observed in acral melanomas in our cohort. Moreover, only 25.9% (7/27) of the cutaneous melanomas with hailstorms showed more than 1 hailstorms (n=2 for all), whereas 72.4% (21/29) of the acral melanomas with hailstorms showed more than 1 hailstorms, with a maximum of five hailstorms per case in 3 cases. This pattern is consistent to what was originally described in Curtin et al. NEJM 2005. We were unable to infer the timing of hailstorms in sun-exposed melanomas as the TCGA does not include sequencing data from independent progression stages of the cases.

We note that it is possible that some acral/mucosal melanomas may have remained despite our filtering since the primary site of most of the metastases and some of the primary melanomas were either unknown or only vaguely described.

- Are some of the acral melanoma amplifications ecDNA? These may explain events that disappear or fluctuate in copy number such as the YAP1 and RAF1 amplification in Figure 4c. FISH may be helpful to confirm.

This is an interesting question, although it reaches beyond the focus of our study on tumor evolution of acral melanoma. To address it at least partially, we performed whole-genome sequencing for the normal, *in situ* and two invasive areas of case 62 with a mean deduplicated sequencing depth of 56x (53-61x for the 4 samples), aligned the reads and applied the tool AmpliconArchitect (Deshpande et al. *Nature Communications*. 2019). We detected two probable ecDNAs in MIS and Invasive A. The first ecDNA contains two segments (**Fig R5A and R5B**), one from 11q at 101825843-102105711 encompassing YAP1, and the other from 3q at 144518784-144825341. The second ecDNA consists of five segments from 3p (**Fig R5C and R5D**), one encompassing RAF1. We have added to the revised manuscript that YAP1 and RAF1 amplicons are indeed likely caused by ecDNAs (**Figure S7**), as the reviewer suspected. However, we consider it less likely that the amplifications occurred early on the evolutionary trunk and then "disappeared" in the invasive B / metastases branch. We did not find any junction-supporting reads in invasive B at all. Considering that the amplifications of YAP1 and RAF1 are expected to convey a selective advantage it seems unlikely that they would disappear without trace. Furthermore, in *in situ* and Invasive A, the flanking regions of the amplicons showed loss of heterozygosity (LOH), while for Invasive B and metastases, these regions retained both parental alleles. The former copy number pattern, i.e., LOH, is not likely the more precursor state. Together, it seems more probable that the ecDNAs arose later during progression, after the MIS / Invasive branch diverged from the common ancestor.

Figure R5. Two extrachromosomal DNAs (ecDNAs) revealed by AmpliconArchitect in MIS and Invasive A of Case 62. A and C) The detailed architecture of the two amplicons, with SV junctions, as denoted by arched lines, indicating circular structures for both. The amplified segments with genomic coordinates indicated were labelled with Greek letters over each segment. B and D) Diagrams illustrating the order and orientation of the amplified segments. For Amplicon 2, the δ and ϵ segments

We also applied AmpliconArchitect to the WGS data of case 110. However, due to the high complexity of the hailstorms within this tumor, we were unable to conclusively determine whether some of the segments formed ecDNA or not.

- Do the APOBEC mutations co-occur with amplifications? Are the APOBEC mutations also early? It may be useful to project COSMIC signatures on the phylogenetic trees. This is sort of done for UV in Figure 3 but does not appear to use COSMIC signatures (see below).

Results of our signature analysis can be found in **Figure S6**. The APOBEC mutation signatures (SBS2/13) were found on the trunk of phylogenetic trees for cases 49, 88, 100, 105, 110 and E, as well as on the branches for cases E and A19T. However, it is likely that our analysis overlooked instances of APOBEC mutations, considering that they constitute only a small fraction of overall mutations and our analyses was limited to coding regions.

Figure R6. A) An example with kataegis within hailstorm common to all tumor areas; **B)** Another example case with kataegis outside of hailstorms and only present in one area, indicating that arose later during progression.

Despite the limited sensitivity of signature analysis, we observed clusters of kataegis, most likely attributable to APOBEC (e.g., PMID: 32025007), falling within hailstorms. As shown in **Figure 1A**, clusters of mutations, mostly C>T mutations, were observed on chr12q of case 101. These mutations were shared across all tumor areas of this case and thus fell on the trunk of phylogenetic tree (**Figure R6A**). Among the 29 cases with hailstorms, 18 had kataegis foci within hailstorms. These hailstorm-associated kataegis foci were found to be located on the trunk of phylogenetic trees in 90% (16/18) of the cases. In addition, we also observed a few scenarios in which kataegis foci resided outside of hailstorms (**Figure R6B**). These hailstorm-independent kataegis foci likely arose later during progression as they tended not to be shared between progression stages.

The finding that the kataegis foci within hailstorms are common to all progression stages of multiple cases further support the clonal nature of hailstorms and the view that they are genomic scars resulting from of genomic alterations that happened in the shared ancestor, early during the evolution of these cancers.

We have revised the manuscript to include this information.

Minor points

- Criteria for hailstorms in the methods are unclear. How exactly is the hypergeometric test applied in each arm or 50 Mbp window to assess CNT density? Specifically, what data is put in the contingency table?

For the hypergeometric test, we considered the following four numbers as inputs: 1) the number of CNTs inside a specific chromosome arm or 50 Mbp window (denoted as x), 2) the total number

of CNTs across the genome (or k), 3) the length of the specific chromosome arm or 50 Mbp window (or m), and 4) the length of the rest of the genome (or n). Thus, the contingency table would look like:

	The specific chromosome arm or 50Mbp window	The rest of the genome
Numbers of CNTs	x	$k - x$
Lengths of Genomic regions (bps)	m	n

We have revised the manuscript to clarify this.

- The text mentions various COSMIC signatures (such as SBS7) but then Figure 3 shows only UV mutations and defined differently (using criteria based on dinucleotides ie "[C/T]C>T or CC>TT"). Which is actually used in the analysis - COSMIC or this definition?

We conducted signature analysis using deconstructSigs and the result is shown in **Figure S6**. Separately, we examined the fraction of [C/T]C>T or CC>TT mutations and presented in **Figure 3**. The reason for separately examining the fraction of [C/T]C>T or CC>TT mutations is that for some of the branches the number of somatic mutations was too low to reliably conduct signature analysis, whereas the fraction of [C/T]C>T or CC>TT mutations could still be informative to infer whether there was UV exposure.

- deconstructSigs in line 535 is misspelled (deconstrucSigs)
Corrected.

Reviewer #2 (Remarks to the Author): Expert in acral melanoma genomics and intratumour heterogeneity

The authors use WES data from FFPE tissues of different countries to explore the genetic evaluation of acral melanoma. The quantity genetic analysis demonstrates the specific characteristics of different stage of acral melanoma. From the evolutionary trajectories of acral melanomas substantially diverge from those of melanomas on sun-exposed skin, where MAP-kinase pathway activation initiates the neoplastic cascade followed by immortalization later. The data demonstrate that punctuated formation of hailstorms, paired with early TERT activation, suggests a unique mutational mechanism underlying the origins of acral melanoma. There are several issue need make clearly.

1. Most of the cases come from East Asia. Does the racial disparities exist? How to avoid this? More and more Chinese acral melanoma patient data have been published and it should be included, at least using as a validation.

Variation of cancer-specific attributes, genetic and others, are an important factor to consider. Epidemiological studies indicate that - contrary to melanomas from sun-exposed skin - the absolute incidence of acral melanoma is comparable across human populations across the globe. To date, genomic studies of acral melanoma have not revealed any significant differences in their genomic alterations (Wang et al and others). In particular, the hailstorms appear to be consistent

feature independent of ethnicity (Curtin, Hayward, Wang etc. see below). Therefore there is no *a priori* evidence suggesting that stratification by ethnicity is a prime concern.

We acknowledge the increasing number of acral melanoma data from Chinese patients in the literature. However, for our analyses of the evolution of acral melanoma we require datasets from cases that include multiple progression stages. We examined whole exome/genome sequencing data sets of acral melanomas in the literature and found that the Farshidfar et al. 2022 Nat Commun study and the Shi et al. 2022 Clinical Cancer Research study contained 67 and 60 Chinese patients, respectively. However, all had only a single tumor area sequenced. The genomic landscapes of acral melanomas from these Chinese patients in these studies were consistent with those observed in other studies. The Zhang et al. 2019 study in the Journal of Investigative Dermatology sequenced multiple primary areas for each of the 7 patients. We downloaded the sequencing data but found the sequencing depths too shallow (< 0.5x for all samples), precluding any detailed analyses. Nevertheless, this study's report of a late-occurring amplicon on chromosome 12 is reminiscent of our observations of late-occurring *CCND1* amplification in case 58 and *MDM2* amplification in case 74.

In response to the reviewer's concern, we have added a comment to the revised manuscript acknowledging the limitation of our study with regard to potential variation among ethnicities.

2. The results and conclusions need more validations at different levels. At genetic level, the WGS and EWS data from the Cell Discovery paper (Cancer Discov.2022 Dec 2;12(12):2856-2879.doi: 10.1158/2159-8290.CD-22-0603) need be cited and make combination analysis. So many RNA data could also be used.

We reviewed the Cancer Discovery study and noticed that, while whole-genome sequencing data were available for 87 acral melanomas, only one sample was sequenced for each tumor. As mentioned above, for our analyses multiple progression stages were required for each case. Nevertheless, we have cited this paper in the discussion section, noting that a finding of this study, i.e., acral melanomas tend to have longer telomere lengths compared to cutaneous melanomas, could be supportive of physical force-induced DNA breaks as the possible mechanism of hailstorm formation. Moreover, the acral melanoma data in this Cancer Discovery paper was already published previously (Newell et al 2020), which we analyzed as part of our recent study (Wang et al 2022). We observed mostly concordant genomic alterations with our dataset, including abundant complex aberrations with high-level amplifications and similar frequencies of driver mutations.

We are aware of the existence of RNA-seq data of acral melanomas from previous studies. However, most of these again were from cases with only one tumor sample sequenced. The focus of our study is on the distinct mutational mechanism as well as the timing in which different driver alterations undergo selection. We do not see how existing RNA-sequencing data could be used to validate or support the main conclusions in our study. Moreover, while we share the reviewer's belief that an open-ended analysis of publicly available RNA-sequencing data could produce interesting observations, such an analysis goes beyond the scope of our current study.

3. Does the different specific anatomic position of acral melanomas (foot, finger, et al) affect the oncogenic events? Genetic data from Nature paper (Nature . 2022 Apr;604(7905):354-361. doi: 10.1038/s41586-022-04584-6.) need be analysed.

This is an interesting question and is related to the reviewer's first point. Similar to ethnicity there is no strong indication that the anatomic site of the primary melanoma affects the genomic

characteristics of acral melanoma. In our previous meta-analysis of 147 acral melanomas (Wang et al. 2022. *Genome Medicine*), we only found subtle differences which will have to be validated independent. Prior copy number studies using comparative genomic hybridization found focused amplifications (commonly residing within hailstorms) throughout all acral melanomas, irrespective of anatomic site (Curtin, Hayward,). So there was also no *a priori* data suggesting that anatomic site maybe an important co-variable regarding the genomic alterations in acral melanoma.

Our cohort is not designed/powerd to identify genetic differences across tumors from different anatomic positions. Among the 37 cases in our cohort, 27 were from foot and the remaining 10 were from the hand (8 from fingers and the other 2 from palms). There were no differences with regard to the main finding of our study (timing of the emergence of hailstorms and *TERT* alterations) between these sites.

The Nature paper the reviewer mentioned focuses on the differences between acral melanomas and melanomas from other parts of the body, rather than the differences among acral melanomas from different acral sites such as foot and finger. The study compared sequencing data of 100 acral melanomas with 839 cutaneous melanomas, and the sequencing was performed through the MSK-IMPACT platform, which is a targeted approach that sequences several hundreds of cancer genes. The underlying data are not publicly available to look for differences in mutation patterns within acral melanomas from different sites. However, we point out that this would not help with our analysis focused on the genomic evolution of acral melanoma.

We added a comment to the discussion stating the limitation of our study to identify any variation among different acral sites.

4. The genetic data get from the FFPE tissues. The results should be validated using fresh acral tissues. Even it is hard to get the TIS, invasion, and metastasis tissues from fresh samples, it could use the freeze section to make sure the quantity and quality of the different samples.

While the DNA from FFPE tissues is more fragmented than DNA from fresh tissue, its quality is fully sufficient for genomic studies on cancer. Collection of fresh tumor tissue is limited to large primary tumors or metastases as all samples need to undergo full histopathological examination to avoid interference with clinical care. FFPE tissues can yield high quality genetic data and have been widely used in previous studies (examples see: Van Allen et al. *Science*, 2015; Liu et al. *Nature Medicine*, 2019). DNA from FFPE tissue is routinely used for next-generation sequencing-based analyses for clinical testing for mutations, copy number and structural rearrangements to make patient management decisions. For our study, we used rigorous quality control checkpoint, and the highly concordant findings across samples from individual patients validate of the quality of the sequencing data and our analyses.

REVIEWERS' COMMENTS

Reviewer #1 (Remarks to the Author):

The authors have adequately addressed all my concerns. I congratulate them on an interesting dataset and study.

Reviewer #2 (Remarks to the Author):

The response letter and the revised manuscript demonstrated my concerns.

Reviewer #3 (Remarks to the Author): Expert in computational cancer genomics and evolution, genome instability, and complex structural variants

I was asked to review specifically the responses to Reviewer 1, but I read the full manuscript and found it very clear and well written, providing important new findings regarding the genetic evolution of acral melanoma.

In my opinion, all the points raised by Reviewer 1 were nicely addressed, and the manuscript was significantly improved, notably by adding whole genome sequencing (WGS) data for representative cases.

My only concern relates to the introduction of a new term, "hailstorms", to define the complex amplification events identified in their cohort. As noted by Reviewer 1, many types of amplifications and complex genomic rearrangements (CGR) were previously defined based on WGS data, and the authors agreed that "hailstorms" could correspond to a "whole exome version" of "amplified chromothripsis" or "tyfonas". They argued that they deliberately chose a new term because they cannot rigorously name them with terms requiring WGS signatures. However, this is confusing to the reader as it seems that "hailstorms" are new biological event. I would refrain from introducing this new term, which is not key to the manuscript, and rather call them "complex amplifications", explaining in the Discussion what they could correspond to. Or, at least, I would suggest the authors to remove this term from the abstract and introduce it carefully in the beginning of the results, clearly indicating that these events likely correspond to previously defined CGR types that cannot be assessed with WES data.

Reviewer #3 (Remarks to the Author): Expert in computational cancer genomics and evolution, genome instability, and complex structural variants

I was asked to review specifically the responses to Reviewer 1, but I read the full manuscript and found it very clear and well written, providing important new findings regarding the genetic evolution of acral melanoma.

In my opinion, all the points raised by Reviewer 1 were nicely addressed, and the manuscript was significantly improved, notably by adding whole genome sequencing (WGS) data for representative cases.

My only concern relates to the introduction of a new term, "hailstorms", to define the complex amplification events identified in their cohort. As noted by Reviewer 1, many types of amplifications and complex genomic rearrangements (CGR) were previously defined based on WGS data, and the authors agreed that "hailstorms" could correspond to a "whole exome version" of "amplified chromothripsis" or "tyfonas". They argued that they deliberately chose a new term because they cannot rigorously name them with terms requiring WGS signatures. However, this is confusing to the reader as it seems that "hailstorms" are new biological event. I would refrain from introducing this new term, which is not key to the manuscript, and rather call them "complex amplifications", explaining in the Discussion what they could correspond to. Or, at least, I would suggest the authors to remove this term from the abstract and introduce it carefully in the beginning of the results, clearly indicating that these events likely correspond to previously defined CGR types that cannot be assessed with WES data.

We thank the reviewer for the positive comments and have removed the term "hailstorms" from the abstract. We have also indicated that these events likely correspond to previously defined "tyfonas" using whole genome sequencing data and have features of "amplified chromothripsis". We opted to continue the use of the term with its specific definition for clarity. Also, considering the additional information on these genomic events provided in our manuscript, namely their punctuated emergence, possible initiation by physical force and re-stabilization by telomere maintenance mechanisms, we consider it possible that they are in fact a distinct biological phenomenon and not necessarily synonymous with the other terms.